# Reduction in the Risk of Peripheral Neuropathy and Lower Decrease in Kidney Function with Metformin, Linagliptin or Their Fixed-Dose Combination Compared to Placebo in Prediabetes: A Randomized Controlled Trial

**DOI:** 10.3390/jcm12052035

**Published:** 2023-03-03

**Authors:** Rafael Gabriel, Nisa Boukichou-Abdelkader, Aleksandra Gilis-Januszewska, Konstantinos Makrilakis, Ricardo Gómez-Huelgas, Zdravko Kamenov, Bernhard Paulweber, Ilhan Satman, Predrag Djordjevic, Abdullah Alkandari, Asimina Mitrakou, Nebojsa Lalic, Jesús Egido, Sebastián Más-Fontao, Jean Henri Calvet, José Carlos Pastor, Jaana Lindström, Marcus Lind, Tania Acosta, Luis Silva, Jaakko Tuomilehto

**Affiliations:** 1Departamento de Salud Internacional, Escuela Nacional de Sanidad, Instituto de Salud Carlos III, 28029 Madrid, Spain; 2World Community for Prevention of Diabetes Foundation (WCPD), 28001 Madrid, Spain; 3Asociación para la Investigación y Prevención de la Diabetes y Enfermedades Cardiovasculares (PREDICOR), 28001 Madrid, Spain; 4EVIDEM CONSULTORES, 28030 Madrid, Spain; 5Department of Endocrinology, Jagiellonian University Medical College, 31-008 Krakow, Poland; 6Medical School, National and Kapodistrian University of Athens, 11527 Athens, Greece; 7Internal Medicine Department, Regional University Hospital of Málaga, Biomedical Research Institute of Málaga (IBIMA), University of Málaga (UMA), 29018 Málaga, Spain; 8Clinic of Endocrinology, University Multi-Profile Hospital for Active Treatment Alexandrovska EAD, Medical University of Sofia, 1431 Sofia, Bulgaria; 9Gemeinnuetzige Salzburger Landeskliniken Betriebsgesellschaft (SALK), 5020 Salzburg, Austria; 10Division of Endocrinology & Metabolism, Department of Internal Medicine, Istanbul University, 34093 Istanbul, Turkey; 11General Hospital Medical System Beograd–MSB, 11010 Belgrade, Serbia; 12Dasman Diabetes Research Institute, Kuwait City 15462, Kuwait; 13Alexandra Hospital, University of Athens, 11635 Athens, Greece; 14Faculty of Medicine, University of Belgrade, 11000 Belgrade, Serbia; 15Renal, Vascular and Diabetes Research Laboratory, Spanish Biomedical Research Centre in Diabetes and Associated Metabolic Disorders (CIBERDEM), Instituto de Investigación Sanitaria Fundación Jiménez Díaz, Universidad Autónoma, 28040 Madrid, Spain; 16IMPETO Medical, 92130 Paris, France; 17Instituto Universitario de Oftalmobiología Aplicada (IOBA), Hospital Clínico Universitario, Universidad de Valladolid, 47011 Valladolid, Spain; 18Department of Public Health and Welfare, Finnish Institute for Health and Welfare, 00271 Helsinki, Finland; 19Department of Molecular and Clinical Medicine, University of Gothenburg, 413 45 Gothenburg, Sweden; 20Department of Medicine, NU-Hospital Group, 451 53 Uddevalla, Sweden; 21Department of Internal Medicine, Sahlgrenska University Hospital, 413 45 Gothenburg, Sweden; 22Department of Public Health, Universidad del Norte, Barranquilla 080001, Colombia; 23Department of Public Health, University of Helsinki, 00014 Helsinki, Finland; 24Diabetes Research Group, King Abdulaziz University, Jeddah 21589, Saudi Arabia

**Keywords:** prediabetes, peripheral neuropathy risk, glomerular filtration, antidiabetic drugs, lifestyle intervention

## Abstract

Objective: To compare the effect of glucose-lowering drugs on peripheral nerve and kidney function in prediabetes. Methods: Multicenter, randomized, placebo-controlled trial in 658 adults with prediabetes treated for 1 year with metformin, linagliptin, their combination or placebo. Endpoints are small fiber peripheral neuropathy (SFPN) risk estimated by foot electrochemical skin conductance (FESC < 70 μSiemens) and estimated glomerular filtration rate (eGFR). Results: Compared to the placebo, the proportion of SFPN was reduced by 25.1% (95% CI:16.3–33.9) with metformin alone, by 17.3% (95% CI 7.4–27.2) with linagliptin alone, and by 19.5% (95% CI 10.1–29.0) with the combination linagliptin/metformin (*p* < 0.0001 for all comparisons). eGFR remained +3.3 mL/min (95% CI: 0.38–6.22) higher with the combination linagliptin/metformin than with the placebo (*p* = 0.03). Fasting plasma glucose (FPG) decreased more with metformin monotherapy −0.3 mmol/L (95%CI: −0.48; 0.12, *p* = 0.0009) and with the combination metformin/linagliptin −0.2 mmol/L (95% CI: −0.37; −0.03) than with the placebo (*p* = 0.0219). Body weight (BW) decreased by −2.0 kg (95% CI: −5.65; −1.65, *p* = 0.0006) with metformin monotherapy, and by −1.9 kg (95% CI: −3.02; −0.97) with the combination metformin/linagliptin as compared to the placebo (*p* = 0.0002). Conclusions: in people with prediabetes, a 1 year treatment with metformin and linagliptin, combined or in monotherapy, was associated with a lower risk of SFPN, and with a lower decrease in eGFR, than treatment with placebo.

## 1. Introduction

Type 2 diabetes (T2D) is a slowly progressive disease characterized by advancing hyperglycemia. Prediabetes, the intermediate stage between normoglycemia and T2D, is a heterogeneous condition with several pathophysiological phenotypes [1]. Progression from prediabetes to T2D ranges between 5–10% per year [2,3,4] and this rate depends on the sub-phenotype of prediabetes and the risk factor profile; for instance, the level of diabetes risk score [2,3,4].

People with prediabetes have an increased risk for macro- and microvascular (namely nephropathy, neuropathy and retinopathy), complications [5,6,7,8,9,10,11]. The Rotterdam study reported that people with prediabetes have generalized microvascular dysfunction and sequelae representing end-organ damage typical of diabetes [12]. Based on such evidence it has been proposed that the definition of prediabetes should not only be centered on glucose abnormalities but should also incorporate microvascular involvement among its criteria [13,14,15].

Lifestyle intervention studies have shown a reduction in the relative risk of diabetes by 40–70% in individuals with prediabetes [13]. The Finnish Diabetes Prevention Study (DPS) and the US Diabetes Prevention Program (DPP) showed how lifestyle intervention, as compared to the control group, was associated with a 58% reduction in the risk of diabetes after approximately three years. The benefits of lifestyle modification in delaying the onset of T2D in people with prediabetes have been confirmed in meta-analyses of clinical trials [16]. However, approximately half of the people with hyperglycemia in the DPP failed to achieve normal glycaemia with lifestyle interventions alone, and about 1/3 of them progressed to clinical T2D in an average of 10 years [3]. Several glucose-lowering drugs such as acarbose, metformin, rosiglitazone, pioglitazone, insulin glargine and liraglutide have also shown some benefits for the prevention of diabetes in people with prediabetes, but this benefit disappeared when the drug therapy was stopped [17,18,19,20,21]. The dipeptidyl peptidase-4 inhibitor (DPP4i) linagliptin has demonstrated its cardiovascular safety in placebo-controlled trials. It also demonstrated non-inferiority using a composite marker for cardiovascular outcome compared to glimepiride. Both cases happened when the drugs were administered as monotherapy in relatively early T2D [22]. Recently, the VERIFY study [23] has reported on the efficacy and safety of combining metformin with the DPP4i vildagliptin, compared with metformin monotherapy, in early–untreated T2D. Evidence of higher efficacy and safety of dual therapy versus monotherapy and/or lifestyle changes in people with prediabetes does not exist. Evidence for the prevention of microvascular complications is also limited in prediabetes. Few prevention trials have evaluated the potential benefits of lifestyle modification or drug treatment for preventing such complications. The Da Qing study in China was the first published trial reporting a significant long-term reduction in the incidence of diabetic retinopathy with lifestyle modification [24]. The evidence regarding the benefit of lifestyle changes or the use of metformin for the prevention of microvascular complications was also inconclusive in the US-DPP [17]. The DPS has reported that lifestyle intervention improved retinopathy status [25]. Experimental studies have reported that linagliptin has a protective effect on the microvasculature of the diabetic retina, most likely due to a combination of neuroprotective and antioxidative beneficial effects [26]. Several clinical trials have also shown that linagliptin may slow down the progression of albuminuria in patients with T2D and renal dysfunction [27,28].

So far, no trial has reported if microvascular function can be better preserved by combining lifestyle interventions with early glucose-lowering drug treatment (multiple or monotherapy) in people with either early diagnosed T2D [29] or with prediabetes [30]. Therefore, the primary objective of the Early Prevention of Diabetes Complications in People with Hyperglycaemia (ePREDICE) trial is to assess the effects of early intensive management of prediabetes on several microvascular parameters in comparison with placebo and lifestyle modification. The primary endpoint was a combination of three microvascular endpoints: retinal and peripheral nerve and kidney functions in adults with prediabetes. In this paper we report 1 year results on the intermediate independent effect of each glucose-lowering drug regimen and placebo (i.e., lifestyle intervention alone) only on the peripheral nerve (sudomotor function) and eGFR.

## 2. Material and Methods

The ePREDICE study design and protocol (including details on the definition and measurement of different variables and outcomes, description of inclusion/exclusion criteria and the recruitment strategy) have been published elsewhere [31]. In sum, the ePREDICE trial is an international, multicenter, randomized, double-blind, parallel-group, placebo-controlled, primary prevention trial initiated by investigators to examine the impact of metformin (Glucophage^®^, Merck KGaA, Darmstadt, Germany), linagliptin (Trajenta^®^ Boehringer Ingelheim, Ingelheim am Rhein, Germany) and a fixed-dose combination of linagliptin/metformin (Jentadueto^®^ Boehringer Ingelheim, Ingelheim am Rhein, Germany) on microvascular parameters compared with matched-placebo. Participants were randomized with equal probability (1:1:1:1) to metformin 850 mg twice a day; linagliptin 5 mg/once a day plus matched-placebo once a day; fixed-dose combination of linagliptin 2.5 mg/metformin 850 mg twice a day; or matched placebo twice a day. Drug treatment was scheduled for 12 months. All randomized participants were also enrolled in a lifestyle intervention program which included 2 individual sessions followed by 12 group sessions of 1.5 hours duration each. Group sessions were repeated every month. The lifestyle intervention program followed the model developed by the European IMAGE Project [32]. All participants attended in the baseline and 12-month follow-up visits for clinical evaluation of microvascular peripheral and kidney function measurements and OGTT assessment. Primary endpoints for this analysis are the 1 year change in foot electrochemical skin conductance (FESC)—measured in μSiemens by SUDOSCAN^®^; participants were classified according to FESC as low risk of small fiber neuropathy (SFN: FESC > 70 µS) and high risk of SFN (FESC < 70 µS; includes moderate risk of SFN, FESC 50–70 µS, and severe risk of SFN: FESC < 50 µS)—as well as 1 year change in kidney function with estimated glomerular filtration rate (eGFR). Secondary endpoints are 1 year changes in fasting plasma glucose (FPG), 2 hour post-challenge plasma glucose (2 h-PG) and body weight (BW). The study protocol did not change since the previous publication [32]. Here we are only reporting baseline characteristics of the study participants and effects of the study intervention on two single independent components of the primary study composite outcome, i.e., the estimated glomerular filtration rate (eGFR) and small fiber peripheral neuropathy (SFPN) risk at 12 months. In addition, we report the effect of the intervention on body weight (BW) and fasting plasma glucose (FPG).

### 2.1. Statistical Analysis

The present analysis is based on the full analysis set (FAS) available for the baseline and 12 month follow-up in the central database by September 26, 2019. In this report we analyze the 1 year changes in the variables of interest in those participants randomized at baseline, who also completed the treatment assigned and attended the 1-year appointment for clinical re-assessment (*n* = 658). For continuous variables, descriptive statistics (mean, SD, percentages and 95% CI) were used. For the comparison of treatment arms at baseline we used a two-sided Mann–Whitney U-test with significance level (*p* < 0.05) for independent samples. We assessed whether each of the 3 active drugs independently (metformin alone, linagliptin alone or the fixed-dose combination of linagliptin/metformin) had a superior effect on the selected variables compared to the placebo. The 1 year changes for paired samples were analyzed using the Wilcoxon test for continuous variables and the McNemar test for categorical variables. In addition, the mean change obtained from the ANCOVA—adjusted for its baseline value—was calculated for the primary outcome variables. The incidence of serious adverse events (SAE) and adverse effects (AE) of special interest to each study drug compared to the placebo are also presented.

### 2.2. Ethical Issues

The study was approved by all local ethic committees of the participating centers and all the National Medicine agencies of the participating countries. All participants received detailed information about the study, including the explanation of their right to withdraw their participation at any time. All participants signed written informed consent forms. All centers followed the European Good Clinical Practice Guidelines and the Declaration of Helsinki as revised in 2008.

## 3. Results

Figure 1 shows the flow-chart of the study population. A total of 1391 potential eligible individuals were screened using a standard 2 h oral glucose tolerance test (OGTT), and 582 (41.8%) of them were excluded: 318 people (22.8%) because they did not meet all the inclusion criteria; 251 (18.0%) declined to sign the informed consent; and 13 people (0.9%) rejected the pre-assigned treatment. In total, 809 participants were fully assessed at baseline and started the assigned drug treatment. After randomization, no demographic differences between the four resulting study groups were observed.

During the first year of the trial, 151 participants (18.6%) withdrew from the study. The discontinuation rates of assigned study drugs were 13.3% (26 participants) in the placebo group; 25.6% (52 participants) in the metformin group; 21.9% (45 participants) in the linagliptin group; and 13.6% (28 participants) in the combined linagliptin/metformin group. After 12 months, 658 participants (81.3%) attended the 1 year follow-up appointment for clinical reassessment. In people who discontinued treatment we did not unmask the assigned drug unless the participant had a medical reason for it. Despite drug withdrawal, we encouraged these participants to continue in the lifestyle intervention program and to attend scheduled clinical appointments. People who discontinued the medication have been excluded from this analysis.

No differences in the variables of interest were observed at baseline between the four study groups, nor among all randomized groups, nor those who completed the 1 year treatment (Table 1).

Table 2 shows the baseline-adjusted mean and proportion differences, and their corresponding 95% confidence intervals (95% CI), between baseline and 1 year follow-up within each study group and compared to the placebo. After 1 year of treatment, the proportion of participants with high risk of SFPN increased by 29.6% in the placebo group, while the increase was significantly lower in the three active-drug groups: 4.6% in the metformin group; 12.4% in the linagliptin group; 10.1% with the combination linagliptin/metformin; and 9.6% when considering the three active-drug groups together (*p* < 0.001 for all comparisons). Compared to the placebo, the proportion of SFPN high-risk estimation by FESC was reduced by 25.1% (95% CI:16.3; 33.9) with metformin alone; by 17.3% (95% CI 7.4; 27.2) with linagliptin alone; by 19.5% (95% CI 10.1; 29.0) with the combination linagliptin/metformin; and by 20.0% (11.5; 28.5) taking together the three active drugs (*p* value <0.0001 for all comparisons). The 1 year kidney function, measured by eGFR-CKD-EPI per 1.73 m^2^, remained significantly higher with the fixed-dose combination linagliptin/metformin (3.3 mL/min/year, 95% CI: 0.37; 6.22), *p* = 0.0270, than with placebo, but not significantly with metformin monotherapy (3.0 mL/min/year: 95% CI; −0.01; 6.01), *p* = 0.0511, nor with linagliptin monotherapy (1.3 mL/min/year; 95% CI: −2.13; 4.73), *p* = 0.4577. After 1 year of treatment, FPG was significantly lower with metformin monotherapy (−0.3 mmol/L, 95%CI: −0.48; −0.12), *p* = 0.0009, with the combination metformin/linagliptin (−0.2 mmol/L, 95% CI: −0.37; −0.03), *p* = 0.0219, and with the three active drugs taken together (−0.2 mmol/L, 95%CI: −0.349; −0.050), *p* = 0.008, than with placebo. However, no difference between linagliptin monotherapy and placebo was observed. No significant reductions were observed between any of the three active drugs and placebo for 2 h-PG. The 1 year reduction in body weight was significantly greater (−2.0 kg/year, 95% CI: −5.65; −1.65), *p* = 0.0006, with metformin monotherapy and with the combination metformin/linagliptin (−1.9 kg/year, 95% CI: −3.02; −0.97), *p* = 0.0002, than with placebo. However, no significant reductions in body weight were observed with linagliptin monotherapy (−0.1 kg/year; 95% CI: 0.15; 0.95; *p* = 0.5822) nor with the three active drugs taken together (−1.5 kg/year; 95% CI: −4.4; 1.4; *p* = 0.316) compared to placebo.

In comparison with the placebo, no significant changes after 1 year of treatment were observed in other cardiometabolic risk factors such as blood pressure, waist circumference, serum triglycerides and HbA1c.

### 3.1. Drug Adherence

In a random subsample of 200 participants (50 patients per study arm), we monitored the participants’ drug adherence during the whole treatment period with the electronic Medication Event Monitoring Systems (MEMS^®^). The compliance with the assigned study medication was considered optimal (95% or more of the days analyzed) by 75 of the patients. No differences in drug compliance were observed between the four study groups during one year of treatment.

### 3.2. Safety Analysis

Only four SAEs, each in four different patients, were reported, none of them related to the study medication as determined by the responsible clinical investigator. The drug treatment was immediately unmasked in these four people. Additionally, the drug treatment was unmasked by local investigators in seven more people for different medical reasons, such as scheduled surgery or acute illness. Nevertheless, participants were asked to resume the assigned study treatment in an open-label fashion after the resolution of the event that required the unmasking.

A total of 52 participants (6.0%) reported a drug-related AE (24 in the metformin group, 18 in the linagliptin/metformin group, 6 in the placebo group and 4 in the linagliptin group). The most frequent AEs were diarrhea (46.2%) and unspecific digestive intolerance (36.5%). These symptoms were more frequent with metformin (22.2%) and with the combination linagliptin/metformin (15.3%). Symptomatic hypoglycemia, clinically relevant hyperamylasemia or acute pancreatitis were not reported during the period analyzed.

## 4. Discussion

Generally speaking, there is a paucity in the scientific literature of interventional studies on the relationship between prediabetes and microvascular complications, specifically to nephropathy and neuropathy, which are the two main focuses of this manuscript. This makes our study a novel one [30]. The ePREDICE trial is an international, investigator-initiated, randomized, placebo-controlled trial aiming at comparing the effects of different glucose-lowering drugs added to lifestyle management with intervention based on lifestyle management alone on the preservation of microvascular function in individuals with prediabetes. In this report, we focus on the effects of three different therapeutic strategies on kidney and sudomotor functions as well as glycemic parameters.

Participants were predominantly middle-aged, Caucasian, female and overweight/obese. The majority were ex-smokers or current smokers and were taking antihypertensive and lipid-lowering drugs. The randomization procedure efficiently generated well-balanced groups in terms of risk stratification.

The number of participants who completed the assigned drug treatment (81.3%; 658/809) can be considered high in comparison with other primary prevention trials combining anti-diabetic drugs and lifestyle modification, where a high proportion of withdrawals usually occur [17]. In our study, the proportion of participants who discontinued the assigned study treatment during the 1 year follow-up did not differ between the four study groups. Regarding kidney function preservation, linagliptin did not produce significant changes in eGFR compared to the placebo at weeks 6, 12, 18 and 24 in the MARLINA-T2D study, a randomized, placebo-controlled, multicenter, Phase IIIb clinical trial. This study suggested that linagliptin may not influence kidney function in patients with T2D within 24 weeks of treatment [27]. SGLT2 inhibitors are known to be effective in preventing kidney function decline with an effect of approximately 0.9 ml per minute per 1.73 m^2^ (95% CI, 0.61 to 1.25) per year in saved renal function compared to placebo in adults with or without T2D who had an estimated glomerular filtration rate (GFR) of 25 to 75 ml per minute per 1.73 m^2^ of body surface area [33]. In the DECLARE-TIMI-58 randomized trial [34], patients generally had good eGFR at baseline, which is the case in our study. The authors analyzed the extent to which dapagliflozin and placebo were associated with a decrease in eGFR in people with T2D and eGFR > 90 ml/min at baseline. The difference between groups was 2 ml/min during a 4-year treatment, i.e., 0.5 ml/min per year of preventive effect in favor of dapagliflozin. From this perspective the effect observed in our study seems relatively good. In 1 year of treatment, the eGFR only decreased by 0.3 mL/min in the metformin-alone group, 0.6 mL/min with the combination metformin/linagliptin and 1.8 mL/min in the linagliptin-alone group compared with a greater decrease of 3.2 mL/min per year in the placebo group. Therefore, it is essential to confirm if the preventive effect will persist over a longer time.

A study conducted in active Finnish workers [35] assessed sudomotor function with FESC. Participants with the lowest fitness level were involved in a 12 month training program with recording of their weekly physical activity and a final fitness level evaluation. Significant differences in BMI as well as waist and body fat were seen according to SUDOSCAN risk score classification. Correlation between the SUDOSCAN risk score and estimated VO2max was r = −0.57, *p* < 0.0001 for women and −0.48, *p* < 0.0001 for men. A significant increase in estimated VO2max in hand and foot ESC and in SUDOSCAN risk score was observed after lifestyle intervention; it was more important in people with the highest weekly activity during the intervention. This was the first study showing that SUDOSCAN could be used to assess cardio-metabolic disease risk status in a working population and to evaluate individual lifestyle interventions. To our knowledge, the ePREDICE trial is the first randomized, controlled trial in prediabetes assessing the effect of lifestyle intervention in combination with glucose-lowering drugs compared with lifestyle modification intervention on peripheral nerve function.

More recently the GRADE study has also reported no differences among the interventions with respect to the development of microvascular outcomes; the mean overall rate (i.e., events per 100 participant-years) of renal impairment was 2.9, and of diabetic peripheral neuropathy, 16.7 [29].

A possible explanation for the small changes in blood glucose observed in our study could be the mix of people with IFG, IGT and IFG + IGT in the study sample. Future analyses should explore whether a stratification by IFG and IGT separately would produce similar results.

Another interesting finding of our study is a greater reduction in body weight observed with metformin monotherapy and with the fixed-dose combination metformin/linagliptin, but not with linagliptin monotherapy, compared to the placebo. The weight loss in the groups containing metformin in our study, approximately 2%, was somewhat higher than what has been reported in other randomized controlled trials. A systematic review and meta-analysis reported an average weight loss of 1.1 kg with metformin used for varying periods [36]. The randomized design of the trial makes it unlikely that the difference favoring intervention groups containing metformin can be explained by a better adherence to lifestyle intervention. However, in the US Diabetes Prevention Program, the metformin group also achieved a similar weight loss of 2.1% after 2 years; remarkably, this lasted for the next 10 years [37].

One-year differences in other cardiovascular risk factors such as blood lipids and blood pressure were non-significant between the active drug groups and placebo. This finding is consistent with other drug trials in prediabetes using similar therapeutic regimens [38,39]. Although the study protocol encouraged the use of antihypertensive and lipid-lowering drugs when necessary, according to the current guideline recommendations [40], we do not have information on the proportion of participants taking these drugs during the course of the study.

Recently the VERIFY study reported that early combination of metformin and DPPIVi drugs in patients with untreated T2D was associated with higher reductions of HbA1c and FPG, both short-term and long-term, than with metformin in monotherapy [23]. Evidence on drug therapy and lifestyle modification combined, compared with lifestyle intervention alone, for the prevention of T2D is scarce [41,42,43,44,45,46,47]. The results of available studies should be interpreted with caution because in general they were small, short intervention time, non-randomized or open-label trials where systematic bias cannot be excluded. In general, these trials do not support the use of pharmacotherapy in combination with lifestyle intervention to lower the risk in individuals with prediabetes.

## 5. Limitations

The e-PREDICE study was challenging to implement because of its complex multinational, non-commercial design using pharmacologic intervention, carried out by independent academic investigators with 2-pill/day requirements of four different combined pharmacologic regimens in asymptomatic people without medical complaints. Despite these challenges, the mean percentage of days with optimal drug compliance (higher than 80% of the prescribed dose) was 75% in the monitored participants. However, in 25% of participants the compliance with the evening dose was lower than 80%. This suboptimal compliance of daily doses may have contributed to the small differences observed between the placebo and active drug groups. Missing data due to participant withdrawal is also an important limitation when interpreting the results of primary prevention trials. However, participants who discontinued drug treatment were asked to maintain the lifestyle recommendations and to remain in the study for future observational follow-up. In the DPP, a similar proportion of participants discontinued metformin during the first six months of treatment, and this proportion remained stable for the next two and five years [17]. In the DPP, the adherence to placebo was consistently higher than adherence to metformin, which contrasts with our study. Using the same definitions than other pharmacological trials conducted in people with prediabetes, we identified a similar number of SEAs and AEs during the treatment period [48]. The number of self-reported gastrointestinal AEs associated with the study medication was slightly higher in the metformin group than in the other study arms, but this was not statistically significant. Similar figures have been reported by other trials in prediabetes [48].

The effects of both lifestyle modification and pharmacologic treatment on diabetes prevention are usually observed after several years of intervention. However, the potential benefits of pharmacological treatment in contrast to lifestyle intervention disappear when glucose-lowering drug therapy is stopped [49,50,51,52].

One year is a short period of time, where fluctuations of blood glucose are common. A longer follow-up is needed before drawing any conclusions on the reduction in diabetes incidence or the regression to normoglycemia. The encouraging results of ePREDICE on risk factors such as body weight and microvascular kidney and peripheral nerve functions can be considered to be of relevance, since the objective was to prevent early microvascular impairment in prediabetes.

## 6. Conclusions

In people with prediabetes, one-year treatment with metformin and linagliptin, either in monotherapy or combination, was associated with a lower risk of small fiber peripheral neuropathy, and with a lower decrease in estimated glomerular filtration rate, than with placebo. In addition, a greater reduction in FPG and body weight was observed with the metformin monotherapy and the fixed-dose combination metformin/linagliptin than with the linagliptin monotherapy or the placebo.

## Figures and Tables

**Figure 1 jcm-12-02035-f001:**
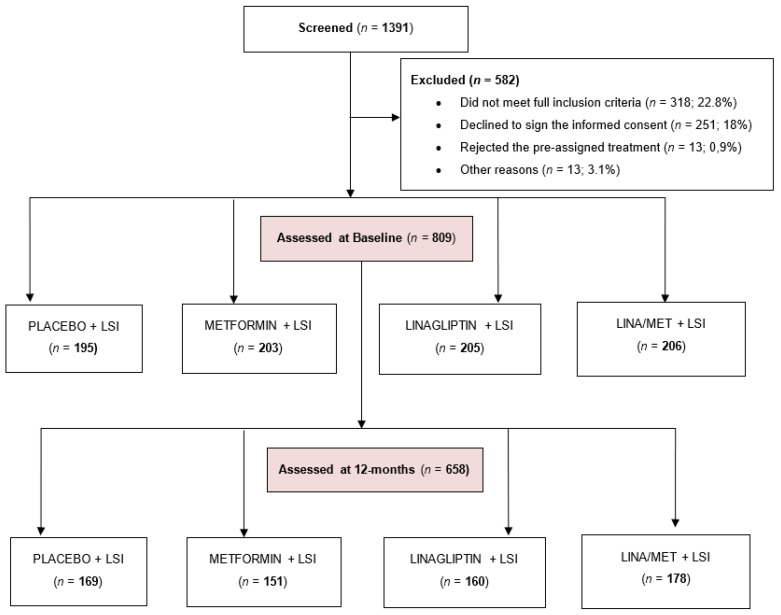
Flow-chart of ePREDICE study population.

**Table 1 jcm-12-02035-t001:** Characteristics of participants at baseline examination by study arm (*n* = 809).

Variables	Placebo (*n* = 195)	Metformin (*n* = 203)	Linagliptin (*n* = 205)	Metformin + Linagliptin (*n* = 206)	*p*-Value
Percent of females	57.7	58.0	58.9	56.8	0.975
Percent of smokers	14.4	15.6	13.2	13.5	0.793
Percent with family history of diabetes	36.5	41.0	37.8	46.4	0.121
Mean age in years (SD)	58.5 (7.5)	58.1 (7.5)	58.1 (7.4)	58.1 (8.0)	0.936
Mean weight in kg (SD)	84.7 (16.9)	83.1 (15.8	84.0 (16.9)	84.4 (16.1)	0.757
Mean BMI in kg/m^2^ (SD)	30.9 (5.1)	30.2 (4.8)	30.8 (5.5)	30.6 (4.6)	0.500
Waist circumference in cm in men (SD)	104.5 (15.7)	105.2 (13.6)	105.1 (12.4)]	107.6 (10.6)	0.791
Waist circumference in women in cm (SD)	101.2 (12.2)	98.9 (11.4)	99.2 (11.9)	98.4 (11.2)	0.791
Systolic blood pressure (SBP) in mmHg (SD)	131 (16.0)	132 (17.1)]	132 (18.0)	132 (17.1)	0.967
Diastolic blood pressure (DBP) in mmHg (SD)	81 (10.9)	82 (10.5)	82 (11.5)	81 (10.2)	0.786
Fasting plasma glucose in mmol/L (SD)	6.3 (0.5)	6.3 (0.5)	6.3 (0.5)	6.4 (0.5)	0.119
Two-hour plasma glucose in mmol/L (SD)	8.1 (1.7)	8.2 (1.7)	8.2 (1.7)	8.1 (1.8)	0.687
Serum total cholesterol (in mmol/L (SD)	5.3 (1.2)	5.3 (1.2)	5.3 (1.1)	5.2 (1.1)	0.652
Serum HDL cholesterol in mmol/L (SD)	1.3 (0.3)	1.3 (0.4)	1.4 (0.4)	1.3 (0.5)	0.885
Serum LDL cholesterol in mmol/L (SD)	3.4 (0.9)	3.3 (1.1)	3.3 (0.9)	3.3 (0.8)	0.755
Serum triglycerides in mmol/L (SD)	1.5 (0.7)	1.5 (1.1)	1.5 (0.7)	1.6 (0.9)	0.474
Percentage of HbA1c (SD)	5.9 (2.6)	5.8 (0.4)	5.8 (0.4)	5.8 (0.4)	0.683
eGFR CKD-EPI in mL/min per 1.73 m^2^ (SD)	94.0 (9.0)	94.3 (9.4)	94.0 (10.0)	94.9 (9.9)	0.729
Mean ESC of feet in µSiemens (SD)	79.2 (10.6)	78.5 (9.8)	79.6 (22.5)	78.0 (9.8)	0.652
Glycemic categories (%)					0.200
Isolated IGT	35.1	33.5	28.3	24.3	
Isolated IFG	37.4	35.8	38.4	39.6	
IGT + IFG combined	27.5	30.7	33.3	36.0	
Percent with hypertension (SBP ≥ 140 mmHg and/or DBP ≥ 90 mmHg or antihypertensive drug use)	58.1	59.0	58.4	60.8	0.940
Percent with hypercholesterolemia (Serum total cholesterol ≥ 200 mg or lipid lowering drug use)	65.3	64.2	66.7	65.8	0.958
Percent with overweight (BMI 25–29 kg/m^2^)	25.2	25.0	31.5	29.7	0.326
Percent with obesity (BMI ≥ 30 kg/m^2^)	52.3	50.0	47.0	53.6	0.535
Percent with abdominal obesity (Men: WC ≥ 94 cm)	41.0	42.9	41.4	46.0	0.856
Percent with abdominal obesity (Women: WC ≥ 88 cm)	59.0	57.1	58.6	54.0	0.856
Percent with diabetic retinopathy (ETDRS > 14)	3.4	4.6	4.9	3.9	1.000
Percent with peripheral neuropathy (feet ESC [µS] <50 or hands ESC [µS] <40)	3.6	6.6	6.8	7.2	0.359
Percent with nephropathy (Albumin: creatinine ratio > 30 mg/dL)	4.5	4.7	8.2	5.0	0.280

BMI: body mass index; HDL: high-density lipoproteins; LDL: low-density lipoproteins; HbA1c: glycated hemoglobin; eGFR CKD-EPI: estimated glomerular filtration rate–chronic kidney disease epidemiology; ESC: electrochemical conductance; IGT: impaired glucose tolerance; IFG: impaired fasting glucose; ETDRS: early treatment diabetes retinopathy study.

**Table 2 jcm-12-02035-t002:** Changes after 1-year treatment by study group and comparisons with placebo in participants who completed the 1-year follow-up examination (*n* = 658).

Variable	Placebo (*n* = 169)	Metformin Monotherapy (*n* = 151)	Linagliptin Monotherapy (*n* = 160)	Metformin/Linagliptin Combination (*n* = 178)
Body weight (kg)				
Mean (SD) baseline	84.9 (17.0)	84.0 (16.8)	84.5 (16.9)	84.0 (17.0)
Mean (SD) year1	83.8 (17.0)	80.9 (16.1)	83.3 (17.3)	80.9 (16.0)
Mean diff. year1-baseline (95% CI)	−1.1 (−1.8; −0.5)	−3.1 (−4.1; −2.2)	−1.2 (−2.0; −0.4)	−3.1 (−3.7; −2.4)
Baseline-adjusted year1 mean (SD)	83.2 (17.0)	81.2 (16.1)	83.1 (17.1)	81.3 (16.6)
Mean difference with placebo (95% CI); *p* value		2.0 (−5.6532; −1.6532); *p* = 0.0006	0.1 (.1521; 0.9523); *p*= 0.5822	1.9 (−3.0291; −0.9709) *p* = 0.0002
Waist circumference in males (cm)				
Mean (SD) baseline	104.8 (11.4)	104.8 (11.5)	105.6 (11.7)	107.7 (12.4)
Mean (SD) year1	105.0 (11.1)	103.3 (11.4)	104.8 (11.8)	104.6 (12.6)
Mean diff. year1-baseline (95% CI)	0.2 (−3.4; 3.9)	−1.5 (−3.9; 0.8)	−0.8 (−2.3; 0.8)	−3.1 (−4.5; −1.7)
Baseline-adjusted year1 mean (SD)	105.6 (11.2)	103.8 (11.3)	104.9 (11.8)	103.5 (12.7)
Mean difference with placebo (95% CI); *p* value		1.8 (−4.2780; 0.6780) *p* = 0.1539	0.7 (−3.1945; 1.7945) *p* = 0.5813	−2.1 (−4.6335; 0.4335) *p* = 0.1039
Waist circumference in females (cm)				
Mean (SD) baseline	101.4 (12.3)	99.8 (11.8)	100.0 (11.9)	98.4 (12.4)
Mean (SD) year1	99.4 (12.7)	96.4 (11.9)	98.5 (12.1)	96.2 (12.8)
Mean diff. year1-baseline (95% CI)	−2.0 (−3.6; −0.4)	−3.4 (−5.0; −1.7)	−1.5 (−3.0; −0.1)	−2.2 (−3.7; −0.8)
Baseline-adjusted year1 mean (SD)	98.1 (12.6)	96.5 (12.2)	98.4 (12.0)	97.4 (12.7)
Mean difference with placebo (95% CI); *p* value		−1.6 (−4.3348; 1.1348) *p* = 0.2506	0.3 (−2.3716 to 2.9716) *p* = 0.8253	−0.7 (−3.3725; 1.9725) *p* = 0.6068
Systolic blood pressure (mmHg)				
Mean (SD) baseline	132 (13.7)	132 (14.0)	132 (14.7)	131 (14.8)
Mean (SD) year1	128 (13.6)	127 (15)	129 (16.1)	128 (14.1)
Mean diff. year1-baseline (95% CI)	−4.2 (−6.5; −1.8)	−4.9 (−7.5; −2.2)	−2.6 (−5.3; −0.0)	−2.9 (−5.2; −0.6)
Baseline-adjusted year1 mean (SD)	128 (13.6)	127 (14.9)	129 (16.2)	128 (13.9)
Mean difference with placebo (95% CI); *p* value		−1.0 (−4.1347; 2.1347) *p* = 0.5307	1.0 (−2.2378; 4.2378) *p* = 0.5439	0.0 (2.9056; 2.9056) *p* = 1.0
Diastolic blood pressure (mmHg)				
Mean (SD) baseline	81 (10.1)	82 (9.8)	81 (10.5)	81 (9,4)
Mean (SD) year1	79 (9.6)	78 (9.0)	79 (9.4)	78 (9.4)
Mean diff. year1-baseline (95% CI)	−1.5 (−3.2; 0.1)	−4.5 (−6.3; −2.7)	−1.3 (−3.1; 0.5)	−2.9 (−4.5; −1.3)
Baseline-adjusted year1 mean (SD)	79 (9.6)	77 (9.3)	79 (9.4)	78 (9.4)
Mean difference with placebo (95% CI); *p* value		−2.0 (−4.0841; 0.0841) *p* = 0.0599	0.0 (−2.0622; 2.0622) *p* = 1.0	1.0 (−3.0064 to 1.0064) *p* = 0.3276
Fasting plasma glucose (mmol/L)				
Mean (SD) baseline	6.3 (0.8)	6.3 (0.8)	6.37 (0.6)	6.4 (0.8)
Mean (SD) year1	6.4 (0.7)	6.1 (0.9)	6.4 (0.8)	6.2 (0.9)
Mean diff. year1-baseline (95% CI)	0.1 (−0.03; 0.2)	−0.2 (−0.3; –0.04)	0.03 (−0.1; 0.1)	−0.2 (−0.3; −0.1)
Baseline-adjusted year1 mean (SD)	6.4 (0.7)	6.1 (0.9)	6.4 (0.8)	6.2 (0.8)
Mean difference with placebo (95% CI); *p* value		0.3 (−0.4764; −0.1236) *p* = 0.0009	0,0 (−0.1628; 0.1628) *p* = 1.0	−0,2 (−0.3709; −0.0291) *p* = 0.0219
2-hour plasma glucose (mmol/L)				
Mean (SD) baseline	8.2 (1.9)	8.3 (2.1)	8.3 (2.2)	8.1 (2.3)
Mean (SD) year1	7.9 (2.2)	8.0 (2.4)	7.6 (2.0)	7.6 (2.3)
Mean diff. year1-baseline (95% CI)	−0.3 (−0.6; 0.03)	−0.3 (−0.7; 0.02)	−0.7 (−1.0; −0.4)	−0.5 (−0.8; −0.2)
Baseline-adjusted year1 mean (SD)	7.9 (2.2)	7.9 (2.4)	7.6 (2.0)	7.7 (2.3)
Mean difference with placebo (95% CI); *p* value		0.0 (−0.5060; 0.5060) *p* = 1.0	−0.3 (−0.7568; 0.1568) *p* = 0.1973	−0.2 (−0.6757 to 0.2757 *p* = 0.4088
Serum triglycerides (mmol/L)				
Mean (SD) baseline	1.49 (0.7)	1.63 (1.1)	1.43 (0.9)	1.53 (0.9)
Mean (SD) year1	1.50 (0.9)	1.60 (1.1)	1.40 (0.6)	1.50 (0.8)
Mean diff. year1-baseline (95% CI)	0.01 (−0.1; 0.1)	0.03 (−0.1; 0.2)	−0.03 (−0.1; 0.1)	−0.03 (−0.2; 0.1)
Baseline-adjusted year1 mean (SD)	1.5 (0.9)	1.6 (1.1)	1.5 (0.6)	1.5 (0.8)
Mean difference with placebo (95% CI); *p* value		0.1 (0.1202; 0.3202) *p* = 0.3722	0.0 (−0.1668; 0.1668) *p* = 1.0	0.0 (−0.1851; 0.1851) *p* = 1.0
HbA1c (%)				
Mean (SD) baseline	6.0 (0.6)	5.9 (0.4)	5.7 (0.4)	5.9 (0.4)
Mean (SD) year1	5.8 (0.4)	5.7 (0.4)	5.6 (0.4)	5.6 (0.4)
Mean diff. year1-baseline (95% CI)	−0.2 (−0.7; 0.3)	−0.2 (−0.3; −0.1)	−0.1 (−0.2; −0.1)	−0.3 (−0.3; −0.2)
Baseline-adjusted year1 mean (SD)	5.70 (0.42)	5.68(0.40)	5.64 (0.43)	5.60 (0.38)
Mean difference with placebo (95% CI); *p* value		0.02 (−0.5001; 0.5001) *p* = 0.9853	0.06 (−0.6001 to 0.5001) *p* = 0.8710	−0.1 (−0.5610; 0.3610) *p* = 0.2318
eGFR CKD-EPI (mL/min per 1.73 m2				
Mean (SD) baseline	93.4 (10.6)	94.3 (14.0)	93.6 (13.6)	94.7 (13.7)
Mean (SD) year1	90.2 (8.9)	93.7 (10.5)	91.8 (15.8)	94.4 (11.4)
Mean diff. year1-baseline (95% CI)	−3.2 (−5.6; −0.8)	−0.6 (−2.0; 0.8)	−1.8 (−4.5; 0.9)	−0.3 (−1.7; 1.1)
Baseline-adjusted year1 mean (SD)	90.6 (15.9)	93.6 (10.3)	91.9 (15.8)	93.9 (11.4)
Mean difference with placebo (95% CI); *p* value		3.0 (−0.0148; 6.0148) *p* = 0.0511	1.3 (−2.1397; 4.7397) *p* = 0.4577	3.3 (0.3780; 6.2220) *p* = 0.0270
Proportion of SFPN high-risk at baseline	26.3	39.4	34.8	38.6
Proportion of SFPN high-risk at 1-year	34.1	41.2	39.1	42.5
Difference (1-year change) in high-risk proportion adjusted by baseline	29.6	4.5	12.3	10.1
Difference 1-year (change) in high-risk proportion compared to placebo (95%CI), *p* value	-	−25.1 (−16.3; −33.9) <0.0001	−17.3 (−7.4; −27.2) <0.0001	−19.5 (−10.1; −29.0) <0.0001

## Data Availability

The clinical trial is still ongoing, patients and data are being managed in a masked fashion. Data cannot be made publicly available since it would compromise confidentiality and might reveal the identity or location of participants. Additionally, public availability of data would be in violation of the Spanish Organic Law 15/1999 of protection of personal data (consolidated text 5/3/2011) and the European Law (EU) 2016/679 from European Parliament and European Council of 27 of April 2016 about Data Protection (RGPD). The data Access Committee of Consejería de Sanidad de la Comunidad de Madrid address: c/Plaza Carlos Trías Bertrán n˚7 (Edif. Sollube) Madrid 28020; (protecciondedatos.sanidad@madrid.org) could consider those requests that do not involve any conflict with these legal regulations. Any acceptable request will be processed and evaluated by the ePREDICE Steering Committee.

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
