# Peer review of "Reduction in the Risk of Peripheral Neuropathy and Lower Decrease in Kidney Function with Metformin, Linagliptin or Their Fixed-Dose Combination Compared to Placebo in Prediabetes: A Randomized Controlled Trial"

_jcm, 2023, doi:10.3390/jcm12052035_

Round 1

Reviewer 1 Report

The manuscript entitled “Reduction in the risk of peripheral neuropathy and preservation of kidney function with metformin, linagliptin or their fixed-dose combination, compared with placebo in prediabetes: a randomized controlled trial” is excellently written and its findings are of great clinical interest. The paper is well organized. The introduction provides a thorough background on the studied matter. The materials and methods are detailly described. The results are presented in a precise manner. The discussion puts the obtained results in a relevant clinical context. In my opinion, the paper should be accepted for publication but I would like to raise some minor concerns: 

- Table 1 has not been cited in the text; additionally, p values for differences in baseline characteristics between study arms should be provided in Table 1

- The full names of the discussed studies should be provided instead of acronyms 

- Please, refrain from using term “prediabetic” 

- The text would benefit from slight language polishing mainly to eliminate several typos 

- The reference list should be revised and made uniform in style

Author Response

- Table 1 is now cited  cited in the text and we have included a new colum with  p values for differences in baseline characteristics between study arms.

- The full names of the discussed studies have been provided in the text 

- We have replaced the  term “prediabetic” for "person or individual with prediabetes" 

- The text has been reviewed and polished by two English speaking  professional correctors, and the  typos eliminated 

- The reference list has been reviewed and several citation corrected using the standard Vancouver citation style.

Reviewer 2 Report

The authors have presented findings from the e-PREDICE international, multicenter, randomized, double-blind, parallel-group, placebo-controlled, primary prevention trial looking at a 1 year follow-up of patients receiving combinations of linagliptin and metformin. The authors do a good job in presenting their observations and have discussed their findings and limitations of the study in the current context. 

However, the authors could improve the presentation of the data to make it easy for readers to understand their key findings. Here are some minor suggestions

1) Table 2 needs proper alignments as it becomes difficult to read otherwise. Highlighting the significant findings within table 2 could also be beneficial to readers (bolding or annotating significant data within the table).

2) Since the authors set out to to test the effects of glucose-lowering drugs on peripheral nerve and kidney function in prediabetic patients, adding a graphical representation of these findings might be useful in better presenting the results and ease of readability.

Author Response

1) Table 2 has been aligned and  significant findings  highlightated as suggested.

2) We think that adding a single graphical representation of eGFR and FESC findings would be difficult to compose because these variables have different measurement scales. We think is better and more readable presenting these results in the table, allowing to show in a more accurate way all figures: means, percentages, 95%CI, p values. .